# Food Insecurity and Child Development: A State-of-the-Art Review

**DOI:** 10.3390/ijerph18178990

**Published:** 2021-08-26

**Authors:** Danielle Gallegos, Areana Eivers, Peter Sondergeld, Cassandra Pattinson

**Affiliations:** 1School of Exercise and Nutrition Sciences, Queensland University of Technology, Kelvin Grove, QLD 4059, Australia; 2Woolworths Centre for Childhood Nutrition Research, Queensland University of Technology, South Brisbane, QLD 4151, Australia; 3School of Psychology and Counselling, Queensland University of Technology, Kelvin Grove, QLD 4059, Australia; a.eivers@qut.edu.au; 4Library Services, Queensland University of Technology, Kelvin Grove, QLD 4059, Australia; p.sondergeld@qut.edu.au; 5Institute for Social Science Research (ISSR), The University of Queensland, Indooroopilly, QLD 4068, Australia; c.pattinson@uq.edu.au

**Keywords:** household food insecurity, child development, socio-ecological

## Abstract

Converging research indicates that household food insecurity impedes children from reaching their full physical, cognitive, and psychosocial potential. This state-of-the-art review examines the last decade of research to: (1) describe the impact of the severity and persistence of food insecurity on child development; (2) use a socio-ecological framework to examine significant proximal and distal factors which may interplay; and (3) outline directions for future research. We conducted a systematic review of six databases of published papers from 2011 to June 2021. The search was limited to high-income countries and children aged from birth to 12 years. From 17,457 papers, 17 studies were included in the final review. Transitioning between food security and food insecurity had a significant and lasting effect on academic/cognitive function and behavior (i.e., externalizing), however less clear relationships were seen for psychosocial outcomes and other behaviors examined (i.e., internalizing). There was significant variation in the measurement and thresholds used to define both food insecurity and child development outcomes. Subsequently, comparisons across studies are difficult. Several future recommendations, including incorporation of socio-ecological factors, is provided. In conclusion, this review supports the link between food insecurity and sub-optimal child development; however, there is an imperative to improve and extend current understanding to ameliorate the causes of food insecurity.

## 1. Introduction

Food and nutrition security is a fundamental human right, and exists “when all people at all times have physical, social, and economic access to food, which is safe and consumed in sufficient quantity and quality to meet their dietary needs and food preferences for a healthy and active life” [1]. It is estimated that nearly two billion people or over one-quarter of the world’s population do not have regular access to a nutritious and sufficient food supply [2]. While prevalence is much lower in high income countries, it remains a persistent and ongoing issue affecting between 8–11% of the populations in countries such as Australia, Canada, Denmark, and the United States of America (US) [3,4,5,6]. The highest prevalence rates in these countries are seen among those living with disadvantage or marginalization [3,7]. Food insecurity has been identified as a powerful stressor for families, with significant negative implications for child health and development; these include impacts on physical, social, cognitive, and behavioral development, independent of poverty [8,9]. As this problem is ongoing and immediate, there is an urgent need to explore the impact of food insecurity on child development, to inform strategies that minimize and alleviate its risks.

To date, three systematic reviews and one meta-analysis have been published examining the associations between household food insecurity and child development [8,9,10,11]. All conclude that food insecurity, independent of economic circumstances, is associated with child development outcomes (cognitive, behavioral, and socio-emotional). The proposed pathways of influence of food insecurity on child development include interactions with maternal mental health, parenting behavior, and household psycho-social stress. None of the previous reviews have focused, however, on the impact of food insecurity severity (from worrying about to running out of food) and persistence over time. The current review is unique in that it specifically investigates the impact of food insecurity severity and persistence on child development outcomes using the socio-ecological model as a guiding framework [12]. This model posits that child development is a dynamic process arising from complex interactions across multiple levels of influence (individual, family, institutions, community, society) that are proximal and distal to the child. The overlaying of this framework will assist in identifying not only risk factors but also the protective resources that can be drawn on to strengthen optimal child development [13,14].

As such, this state-of-the-art review will outline the past decade of research to: (1) examine the impact of the severity and persistence of food insecurity on child social, emotional, cognitive, and behavioral development; (2) utilize the socio-ecological model as a framework to examine the factors which may be protective or exacerbate the effect of food insecurity on child development; and (3) outline key directions for future research on food insecurity.

### Definition and Classification of Food Insecurity

Food and nutrition security is underpinned by six dimensions: (1) availability—food of sufficient nutritional quality is able to be grown, bartered or purchased; (2) accessibility—social, economic and physical access to food; (3) utilization—food is able to be used physiologically and there are resources to transform food into meals; (4) stability—that all these elements are stable irrespective of household, civil unrest, or weather conditions; (5) agency—people can choose what they eat and how it is produced with freedom and dignity; and (6) sustainability—indicating long term measures that protect human and environmental health [15]. Food insecurity occurs when one or more of these dimensions are compromised. Food insecurity experiences are most commonly measured at the household level and generally reported by the primary caregiver.

One challenge to the conceptualization of food insecurity is that it may differ in both persistence and severity, with potentially differing consequences for child development. A reason for this variability is that economic disadvantage is dynamic. Households may move in and out of poverty, or at times have greater access to supporting resources than at other times. The chronicity and cyclical nature of disadvantage are, thus, potential moderators of long-term child development outcomes [16,17] and are therefore, key variables of interest within the current review.

## 2. Materials and Methods

The Cochrane Collaboration [18] and Centre for Reviews and Dissemination [19] guidelines were used in the development of this review. We report findings per the Preferred Reporting Items for Systematic Reviews and Meta-Analyses (PRISMA) Statement [20]. We searched MEDLINE (via EBSCOhost), ProQuest (Education; Health & Medical; Nursing & Allied Health; Psychology; Social Science; Dissertations & Theses Global), PsycINFO (via EBSCOhost), SCOPUS, and Web of Science Core Collection for empirical research on links between food insecurity and child development from 2011 to June 2021. We included only full text, English language, peer reviewed publications. Table 1 summarizes eligibility criteria.

We searched the Cochrane Library to identify primary studies in relevant systematic reviews. No additional records were identified via hand-searching the reference lists of records meeting eligibility criteria. We adapted search terms and search syntax for each database (see Appendix A). Papers were limited to ones published within the last decade (2011–2021) and to those including children from birth to 12 years only.

### Risk of Bias

From 17,457 studies screened, 17 met the final inclusion and exclusion criteria. Two assessors determined the inclusion of papers, extracted data and evaluated the quality of each of the studies using the National Heart, Lung and Blood Institute’s Quality Assessment Tool for Observational Cohort and Cross-Sectional Studies [21]. This tool assesses quality across sixteen criteria. Any conflicts were discussed and finalized by the group. Just under half (47%) of the studies were rated as being of good quality, with the remaining papers rated as fair (see Appendix A). Figure 1 depicts the study selection process.

## 3. Results

### 3.1. Measurement of Food Insecurity Severity and Persistence

All but two of the 17 studies identified used the United States-Household Food Security Survey Module (US-HFSSM; Table 2). This tool distinguishes between households which are food secure and food insecure. Households are categorized as food secure if they have high food security (no anxiety, consistently able to access food) or marginal food security (some anxiety about accessing adequate food but no changes to food intake). Households are categorized as food insecure if they have low food security (where the quality of food is compromised but quantity and eating patterns are not altered); or very low food security (where some or all members of the household had disrupted eating patterns and this reduced the quantity of food consumed) [22]. In addition to being able to determine a food security status, the tool can provide a continuous variable where higher scores are indicative of more severe food insecurity. Furthermore, the US-HFSSM can be used to distinguish between food security among adults (the first ten questions) and among children (the remaining eight questions) in a household. There are difficulties, however, when child food insecurity is measured in relation to households as measures do not necessarily capture food security status for all children in the household, with younger children often protected by adults over older children [23].

Fifteen (82%) studies reviewed here used the 18-item or 10-item US-HFSSM measure. Of the remaining two studies, one study used the two-question screener from the US-HFSSM [24] and the other used a four question screener that had been previously validated [25]. All studies categorized households as food secure or food insecure; three studies also examined food insecurity as a continuous variable [26,27,28].

Nine (53%) of the 17 studies reviewed, dichotomized the HFSSM scale into food secure versus food insecure, thereby not distinguishing severity of household food insecurity. One of these studies included those with marginal food security in the food insecure category [29], which is at odds with recommended practice [22] and makes comparisons between studies problematic. Six studies (35%) did distinguish food insecurity severity. Four of these studies did so by comparing the trichotomous outcomes of food secure, marginally food secure, and food insecure [24,26,30,31]. This may be important as there is converging recognition that parental anxiety or worry about food, regardless of objective food security status, may impact on child development through indirect mechanisms such as parenting and home environment [30]. The remaining two studies distinguished severity within the food insecure category, that is, between low and very low food security [26,31]. Nagata and colleagues [26] examined continuous and categorical values of food insecure, with food secure versus marginally food secure, low, and very low food security. Given that very low food security is associated with compromised quality and quantity of food, there are physiological implications for child development with this level of food insecurity [32,33]. In high income countries the number of households with very low food security is often low (between 3–5%) [31] and, is frequently not able to be analyzed separately.

Two studies used the US-HFSSM to distinguish the experiences of food insecurity for adults from that of children in the same household [27,34]. Given that children can be protected in food insecure households by adults, determining whether children are experiencing food insecurity (albeit based on caregiver report) provides a more nuanced understanding of the impact of food insecurity on child development.

Persistence/trajectory of food insecurity was measured in seven (41%) longitudinal studies [25,28,29,31,35,36,37,38]. These papers highlight a growing understanding of the impact of cycling through transient phases of food insecurity over time, as well as the impact of persistent food insecurity at different stages of child development. Transient phases of food insecurity may indicate precarious or chaotic environments (characterized by uncertainty, frequent moving, and lack of routines), which have been linked to poorer child development outcomes [39,40]. Only one study examined the combined impact of severity (food secure, marginal food security and food insecurity, which included both low and very low food security) and persistence across two time points of development (Table 2) [37].

**Table 2 ijerph-18-08990-t002:** Food security measurement tools and characteristics used in identified studies.

Author-Year Country	Severity Y/N	Measurement *	Persistence Y/N	Measurement Notes
Tool used: 18 item US-HFSSM				
Black 2012 USA [41]	N	FS vs. FI	N	
Hobbs 2018 USA [42]	N	FS vs. FI	N	
Howard 2011 USA [29]	N	FS vs. FI (MFS + LFS + VLFS)	Y	Persistent FS = FS 1st to 5th Grade Persistent FI = FI 1st to 5th Grade Contemporaneous FS FS_first-time-_Grade 3; FS_first-time-_Grade 5; FS_second-time-_Grade 5 FI_first-time-_Grade 3; FI_first-time-_Grade 5; FI_second-time-_Grade 5
Huang 2016 USA [35]	N	FS vs. FI	Y	Persistent FS across three waves FI in one of three (Kindergarten) FI in two of three waves (Kindergarten, Grade 3) Persistent FI across three waves (Kindergarten, Grade 3, Grade 5) Patterns of FI kindergarten and third grade; kindergarten and fifth grade; third and fifth grades
Ramsey 2011 Australia [43]	N	FS vs. FI	N	
Huang 2018 USA [44]	N		N	Used wave 4 of ECLS-K only
Drennen 2019 USA [34]	N	FS vs. FI Adult ^#^ FS Child FS Adult FI Child FS Adult FI Child FI	N	
Kimbro 2015 USA [36]	N	FS vs. FI	Y	Families classified into four groups: FS at both waves; became FS (FI in kindergarten and FS in first grade) became FI (FS in kindergarten and FI in first grade); FI at both waves
Jackson 2018 USA [28]	Y	FS vs. FI (status) Continuous variable (index)	Y	FS = FS at Wave 1 and Wave Transient FI = FI at either Wave 1 or Wave 2 Persistent FI = FI at both Wave 1 and Wave 2
King 2018 USA [27]	Y	FS vs. FI Continuous variable Household (0–18) vs. Child (0–8) Adult (0–10) vs. Child (0–8)	N	
Cook 2013 USA [45]	Y	HFS vs. MFS vs. FI	N	
Grineski 2018 USA [37]	Y	HFS vs. MFS vs. FI	Y	PersistFI: FI at TI and T2 PersistMFI: MFS at T1 and T2 EmergFI: FS at T1 and FI at T2 EmergMFI: FS at TI and MFS at T2 DeepFI: MFS at T1 and FI at T2 RemitFI: FI at TI and FS at T2 RemitMFI: MFS at T1 and FS at T2 AttenFI: FI at T1 and MFS at T2 AvertFI: FS at T1 and T2
Johnson 2017 USA [31]	Y	FS vs. FI LFS vs. VLFS	Y	Any household FI at any one wave Any household FI at any two waves Any household FI at all three waves
Nagata 2018 USA [26]	Y	FS vs. MFS vs. LFS vs. VLFS Continuous variable used in regression	N	
Gee 2018 USA [38]	N	FS vs. FI	Y	Recurrent FI—FI at kindergarten and first grade Transient FI—FI at kindergarten wave only
Other tools used				
Encinger 2020 USA [24]	N	Used two-item screener from US-HFSSM FS vs. MFS	N	
Melichor 2012 Canada [25]	N	Used four-item tool loosely based on US-HFSSM	Y	Only one question asked at Wave 1 All questions asked at Wave 4 FI at 1.5 y and 4.5 y FI at 1.5 y and FS at 4.5 y FS at 1.5 y and FI at 4.5 y

Y = Yes; N = No. FS = Food secure; MFS = Marginal food security; FI = Food Insecure; LFS = Low Food Security; VLFS = Very Low Food Security; T1 Time 1; T2 Time 2. * FS as per USDA guidelines, i.e., includes <3 affirmative responses unless otherwise stated (FS + MFS); FI ≥ 3 responses (LFS + VLFS). ^#^ Responses to 10 question adult module to give household FS status; Responses to eight-question child module to give child food security status within household.

### 3.2. Measurement of Child Development

In line with the recent review by de Oliveira et al. [9], the current review found that the measures used to assess child development (see Table 3) varied widely across studies and included a mix of non-standardized single items, summed multiple single-item responses (to get an overall functioning score), and standardized tests [9]. No study provided a rationale for using a measure, even when other validated and more commonly used scales existed. This is problematic as it limits comparability across studies and cohorts. More consistent measures and use of standardized measures are vital as is ensuring context generalizability outside the US. It should be noted that 10 (60%) of the reviewed papers were based on two large cohorts from the United States—the Early Childhood Longitudinal (ECLS) Birth and Kindergarten cohorts and the Fragile Families and Child Wellbeing Study (FFCWS)—where the choice of tools was pre-selected.

### 3.3. Food Insecurity and Child Development Outcomes

Academic/cognitive outcomes: Seven studies investigated the association between food insecurity and academic outcomes/cognitive functioning (Table 3). None investigated the impact of severity of food insecurity on cognitive outcomes; however, five of the seven studies examined the impact of food insecurity persistence.

There were two cross-sectional investigations with mixed findings. Hobbs and King [42] indicated that, compared to children in FS households, children in food insecure households had lower scores on measures of both vocabulary and letter-word recognition, but these effects were different for children in different ability percentiles (Table 3). Huang and colleagues [44] reported that, after adjusting for immigrant protective and risk factors, there were no significant differences in reading or math scores according to food security status.

Of the five longitudinal studies, four reported a significant negative effect of food insecurity persistence on academic/cognitive outcomes. Two studies [29,31] found that both transient and persistent food insecurity were associated with decreased approach to learning [29,31] and reading and math scores [31]. An additional study reported that persistent, but not transient food insecurity was associated with decreases in reading scores [38]. Similarly, Grineski et al. found that only children in households who transitioned from marginally food secure to food insecure (deepening food insecurity) had lower math and working memory scores [37]. Finally, Kimbro and Denney [36] found no associations between either persistent or transient food insecurity and academic outcomes (reading, math or science) across two time points.

Behavior: The effect of food insecurity on behavior (externalizing, internalizing, self-control, self-regulation, general conduct) has had considerable attention over the past decade with 12 studies specifically examining this association. Five were cross-sectional and eight were longitudinal (Table 3). Three of four cross-sectional studies reported positive associations between food insecurity and behavioral problems [24,43,44]. Hobbs and King [42] reported that this effect was greatest in those children who had higher behavioral problems to begin with. Encinger and colleagues [24] found that marginal FS was indirectly associated with poorer self-regulation, mediated through parenting stress. Nagata et al., however, found no direct association between food insecurity and behavior problems [26].

Eight papers examined the effect of food insecurity persistence on behavior and the results were mixed. Four papers examined self-control, all finding significant negative associations with food insecurity [28,29,31,38]. The association was particularly marked where there were transitions into and out of food insecurity, indicating that some level of uncertainty regarding food security within a household may impact child self-control.

Four of the eight longitudinal studies examined the association between internalizing and externalizing behaviors and food insecurity persistence [27,31,37,38]; one study investigated externalizing behavior outcomes only [29]. Two papers reported that emerging food insecurity (food secure at Time 1 moving to food insecure at Time 2) was associated with increased externalizing behavior [36,37]; this finding was replicated by Huang et al. [35], but for boys only. Grineski et al. [37] reported a significant positive association between persistent food insecurity (food insecurity at both time points) and externalizing behaviors. Only two studies found an association between food insecurity persistence and internalizing behaviors [35,36]. Kimbro and Denney indicated a significant positive effect for persistent food insecurity on internalizing behaviors; Huang et al. found a significant positive association between emerging food insecurity and internalizing behavior but, again, for boys only [36]. King [27] found increasing internalizing behaviors in children in households where adults only were food insecure; and increasing externalizing behaviors in children in households where both adults and children were food insecure.

Among the remaining longitudinal studies, a study by Johnson and Markowitz [31] found that food insecurity at any earlier time point was associated with increased hyperactivity and conduct problems in kindergarten. Another study, by Melichor and colleagues, found no longitudinal association between food insecure and hyperactivity and inattention, aggression, or depression [25].

Taken together, these results indicate that food insecurity persistence may differentially affect behavior in children when experienced at different times in their development. Shorter and more transient forms of food insecurity were associated with increased externalizing behaviors, while more persistent food insecurity was associated with internalizing and self-control behavioral issues. Results are mixed, however, and further analysis is needed to disentangle these effects.

Development: Three cross-sectional studies examined food insecurity and developmental concerns (Table 3): all reported that food insecurity was associated with increased developmental concerns reported by parents using the Parents’ Evaluation of Developmental Status (PEDS). These studies each controlled for critical child (birth weight, feeding) and caregiver characteristics (age, education, employment, and marital status) [30,36,42]. Both marginal food security and food insecurity were associated with increased developmental concerns [45].

Psychosocial: Four studies assessed the associations between food insecurity and psychosocial outcomes using a variety of measures, with few recognized and standardized measures being used. Potentially due to this, the patterns of these findings were mixed. Two studies were cross-sectional and two were longitudinal. In a cross-sectional analysis, Nagata et al. [26] reported that after adjusting for child, maternal, and household factors, on all five of the Child Behavior Checklist subdomains, experiencing food insecurity was only significantly associated with declines in pervasive development. Cook et al. [45] found that food insecurity but not marginal food security was associated with decreased odds of the child having “well child” status compared with children in food secure households.

Of the two longitudinal studies, Howard [29] reported that children who transitioned from food insecure in the first grade to food secure in third grade had lower social skills scores, an effect that was significant overall, in boys, and trending towards significant in girls. However, it is noted that there were no other significant associations between social skill scores and food insecurity persistence found, including in those who became food insecure in third grade or those experiencing any food insecurity by fifth grade. Grineski and colleagues [37] found that remitting marginal food insecure (marginally food secure at kindergarten and moved to food insecure at grade 1) and persisting marginal food security (marginally food secure at both kindergarten and grade 1) were associated with declines in teacher-rated interpersonal skills, even after controlling for child and school factors. In combination, these longitudinal studies suggest that transitioning between food security and food insecurity matters, especially in the early years. Furthermore, the results from Grineski et al. [37] suggest that the effects of even marginal food security may impact on children’s interpersonal skills and development, even after food insecurity is no longer a significant household problem.

**Table 3 ijerph-18-08990-t003:** Summary of food security and child development outcomes.

Author-Year (Study)	N	Mean Age (Years)	Control Variables ^#^	Outcome	Findings (Adjusted for Covariates Where Applicable)
Academic/Cognitive					
Cross-sectional					
Hobbs 2018 (FFCW) [42]	1684	~5	Child (birthweight, health, asthma); parent (race, education, relationship, employment, nativity, mother age at birth +3, depression); household (income, number of children, material hardship, social support, parenting stress, parent relationship quality)	Receptive vocabulary and academic readiness; cognitive development (letter-word identification); Categorised children using quartile regression into 4 academic/behaviour outcome percentiles (low–high); 10th, 25th, 75th, 90th.	↓ vocab scores for children between 50th and 90th percentiles; ↓ cognitive development for children in the 10th percentile Boys: ↓ vocab scores for 75th percentile Girls: ↓ vocab scores for 75th and 90th percentiles
Huang 2018 [44] (ECLS-B)	8900	~4.5	Parent immigrant status, family structure, birthweight, maternal depression, socioeconomic status (family income + maternal education), public assistance receipt, child health insurance, household size, primary language at home, years spent in USA, race	Reading and math scores	NS differences in reading or math assessment scores between FI immigrant and FS immigrant, FS US and FI US after immigrant protective and risk factors taken into considerations Differences in scores between immigrant and US children is explained by socioeconomic factors.
Longitudinal					
Gee 2018 [38] (ECLS-K)	1040	5.6	Child (age, sex, disability status, number of siblings, race, health status, home language, free or subsidized lunch) School: ^#^ children qualifying for free lunch, % minorities Other: material hardship (financial hardship, residential mobility, medical care), parental depression, parenting stress, parental warmth and investment, socioeconomic status, marital status, employment status	Reading score	Recurrent FI (RFI) initially assoc. with ↓ reading score but over two years converged with transient FI (TFI) so no difference between RFI and TFI groups at later ages
Grineski 2018 [37] (ECLS-K)	11,958	7.1 (SD = 0.4)	Child (age, sex, race, birthweight, type of care, kindergarten assessment); parent (family structure, nativity, mother age at birth, depression, health); household (size, SES); school (teacher turnover, teacher absence, bullying +6)	Academic outcomes: reading, science, math. Cognitive function: working memory, cognitive flexibility	Deepening FI →↓ math scores and working memory
Howard 2011 [29] (ECLS-K)	4710	1st grade ^	Child-level (age, sex, developmental disability +9) household (^#^ of siblings, structure, income) parental education +8	Approaches to learning	Subject-specific model ^b^ Transition from FI in the first grade to FS in the third grade assoc. with ↓approaches to learning for girls and boys respectively
Johnson 2017 [31] (ECLS-B)	2200–3700	~5	Child (age, sex, race, kindergarten entry +5), parent (education, English fluency, employment +8) and household characteristics (family structure, urbanicity +3), maternal depression	Reading, math and approach to learning	Intensity of household FI at any wave predicting outcomes in kindergarten: FI at any one or two waves assoc. with ↓ reading and math scores in kindergarten FI at any of the waves assoc. with reduced approaches to learning in kindergarten
Kimbro 2015 [36] (ECLS-K)	6300	6.1	Child (age, sex, low birthweight) and family characteristics (SES, ethnicity of parents +3 others). Accounted for school fixed effects	Academic achievement in reading, math, and science	No assoc. with academic outcomes
Behavior *					
Cross-sectional					
Encinger 2020 [24]	249	4.5 (SD = 0.5)	Child (age, sex), family structure (one or two parent), caregiver (education, age at birth), parenting stress, parenting distress	Self-regulation	Marginal FS no direct association with self-regulation Marginal FS indirect effect on self-regulation through parenting stress
Hobbs 2018 [42] (FFCW)	2046	~5	Child (birthweight, health, asthma); parent (race, education, relationship, employment, nativity, mother age at birth +3, depression); household (income, number of children, material hardship, social support, parenting stress, parent relationship quality)	Internalizing, externalizing behaviors	↑ internalizing and externalizing behaviors with greater effects in those in the higher percentiles Boys: ↑ externalizing in 50th and 90th percentiles; ↑ internalizing all percentiles Girls: ↑ externalizing and internalizing behaviors with effects greater for those in higher percentiles
Nagata 2018 [26]	168	~5	Child (sex); maternal (ethnicity, education, marital status, age, mental health status, depression); household (food stamps, SNAP participation)	Anxiety problems, affective score, attention deficit/hyperactivity, oppositional defiant problems	Non-significant effect of FS on affective problems after adjusting for maternal depression No association with oppositional defiant, anxiety, or hyperactivity
Ramsey 2011 [43]	182	3–17	Household income	Behavioral problems—overall difficulties score	↑ in borderline or atypical emotional and behavioral problems
Longitudinal					
Grineski 2018 [37] (ECLS-K)	11,958	7.1 (SD = 0.4)	Child (age, sex, race, birthweight, type of care, kindergarten assessment); parent (family structure, nativity, mother age, depression, health); household (size, SES); school (teacher turnover, teacher absence, bullying +6)	Internalizing, externalizing behaviors, self-control	Persisting marginal, remitting marginal FI, deepening FI →↓ self-control; emerging and persisting FI ↑ externalizing behaviors. No significant effect on internalizing behaviors
Howard 2011 [29] (ECLS-K)	4710	1st grade ^	Child-level (age, sex, developmental disability +9) household (^#^ of siblings, structure, income) parental education +8	Externalizing behaviors, Self-control	Subject-specific model ^b^ Transition from FI in the 1st grade to FS in the 3rd grade assoc. with ↓ self-control for girls and boys respectively
Huang 2016 [35] (ECLS-K)	7348	5.7 (SD = 0.4)	Child (sex, age, race, +9), maternal (age, age at child birth, employment, +4), and household characteristics (number of children, household size, income, +5); parental warmth, parenting stress, parental depression	Externalizing and internalizing behaviors	Overall, long-term patterns of FI not assoc. with changes in behavior For boys: Transitioning to FI at grade 3 was assoc. with ↑ externalizing and internalizing behaviors
Jackson 2018 [28] (FFCW)	2977–3252	9.3 (SD = 0.4)	Child (age, race, sex); mother (education, low self-control, depression, involvement); household (income, public assistance)	Self-control	FI index (continuous) and FI status associated with ↓ self-control and predictive of early delinquency at waves 3 and 4 Transient FI and persistent FI associated with ↓ self-control and greater involvement in early delinquency
Johnson 2017 ^b^ [31] (ECLS-B)	2200–3700	~5	child (age, sex, race, kindergarten entry +5), parent (education, English fluency, employment +8) and household characteristics (family structure, urbanicity +3); maternal depression	Hyperactivity and conduct problems	Intensity of household FI at any wave predicting outcomes in kindergarten: FI at any of the waves was assoc. with ↑ hyperactivity in kindergarten FI at any one or two waves was assoc. with ↑ conduct problems in kindergarten
Kimbro 2015 [36] (ECLS-K)	6300	6.1	Child (sex, age, race, low birth weight), family characteristics (ethnicity, maternal age, SES, +2)	Behavioral; self-control, interpersonal skills, externalizing behaviors, and internalizing behaviors.	Children who transitioned to FI had ↓ interpersonal skills, self-control, and ↑ externalizing behaviors Children who were persistently FI had ↑ internalizing behaviors
King 2018 [27] (FFCW)	2044	~5	Child (health, asthma); maternal relationship with father, employment, SNAP participation, depression +4); household (income, poverty, material hardship, social support, parent relationship quality, parenting warmth, parenting responsibilities, parenting stress, parental alcohol or drug abuse, mother victim of violence)	Internalizing, externalizing behaviors	↑ internalizing and externalizing behaviors; no change with maternal depression, slightly attenuated when take into consideration parenting stress, ↑ internalizing behaviors with households with adult FI ↑ externalizing behaviors with households with both adult and child FI
Melichor 2012 [25] (QLSCD)	1682	8	Child (sex, immigrant status) and family characteristics (family structure, age at child birth, income, +10); parental depression, family functioning	Depression/anxiety Aggression Hyperactivity/inattention	No assoc. with aggression No assoc. with depression/anxiety ↑hyperactivity/inattention No assoc. with hyperactivity between ages 4.5 and 8 years, when adjusting for hyperactivity/inattention at 1.5 years No sex differences
Development (Cross-sectional Only)					
Black 2012 [41]	26,950	<3	Caregiver’s age, educational level, race, country of birth, marital status, employment, depressive symptoms, child’s age, breastfed, and low birth weight.	Developmental risk (1 or more concerns on the PEDS)	↑developmental risk
Cook 2013 [45]	41,515	<4	Research site, mothers’ race, foreign-born status, marital status, education level, depressive symptoms, employment status, age, breastfed, public assistance (TANF, low-income energy assistance)	PEDS (1 important concern, 2 or more important concerns).	Both marginal FS and FI assoc. with ↑ developmental concerns on both 1 concern and 2 or more concerns
Drennen 2019 [34]	28,184	1.5 (SD = 1.1)	Research site, mother’s age, education, race/ethnicity, employment; child’s age, birth weight; food assistance program participation	Development risk (2 or more concerns on PEDS)	FI associated with increased developmental risk at all ages except 25–36 months Odds of developmental risk higher for the Household FI/Child FI in the 0–12-month group and both Household FI/Child FI and Household FI/Child FS groups in older children
Psychosocial					
Cross-sectional					
Cook 2013 [45]	41,515	<4	Research site, mothers’ race, foreign-born status, marital status, education level, employment status, age, breastfed	Composite indicator Child well being; free from adverse conditions labelled ‘well child’	FI assoc. with ↓ odds of having ‘well child’ status Marginal FS not assoc. with “well child” status
Nagata 2018 [26]	168	~5	Child (sex); maternal (ethnicity, education, marital status, age, mental health status, depression); household (foods stamps, SNAP participation)	Pervasive development	↓ pervasive development in unadjusted and adjusted models
Longitudinal					
Grineski 2018 [37] (ECLS-K)	11,958	7.1 (SD = 0.4)	Child (age, sex, race, birthweight, type of care, kindergarten assessment); parent (family structure, nativity, mother age, depression, health); household (size, SES); school (teacher turnover, teacher absence, bullying +6)	Interpersonal skills	Persistent marginal FI and remitting marginal FI were associated with ↓ interpersonal skills
Howard 2011 [29]	4710	1st grade ^	Child-level (age, sex, developmental disability +9) household (^#^ of siblings, structure, income) parental education +8	Social Skills Ratings Scale ^a^ as a composite measure, interpersonal relationships,	Subject-specific model ^b^ Transition from FI in the 1st grade to FS in the 3rd grade assoc. with ↓social skills scores

Note: FI: Food Insecure; FS: Food Secure; NR: Not Reported; PEDS: Parents’ Evaluation of Developmental Status; ↑—increased; ↓—decreased. Studies: ECLS-B and –K: Early Childhood Longitudinal Study, Birth and Kindergarten cohorts; FFCW: Fragile Families and Child Wellbeing Study; QLSCD: Quebec Longitudinal Study of Child Development. ^#^ Control variables are presented in order of appearance in the respective papers, for ease and readability we have tried to present a selection of the key variables controlled for in the analysis. The number of additional variables that were controlled in the analyses are presented with the “+” symbol. ^ Baseline Cohort reported in this instance. ^a^ This table only reports the overall social skills score, please refer to the original manuscript for full report of findings. ^b^ The subject-specific model represents the most conservative estimate of effects, please refer to the original manuscript for full report of findings. * anxiety and depression were classified under behavior as they are often categorized under internalizing behavior.

## 4. Discussion

### 4.1. Mechanisms of How Food Insecurity Impacts on Child Development

Food insecurity has been linked to adverse child development through multiple mechanisms, including decreased quantity of food, compromised food quality, and heightened stress and anxiety associated with finding food [46,47]. A decrease in quantity of food, where children skip or have smaller meals, or potentially changes in the quality of food provided (that is, cheaper nutrient-poor, energy-dense alternatives over nutritious meals) may result in inadequate consumption of required nutrients. For instance, sub-optimal energy, protein, and micronutrient intake in the first five years of life can limit neural plasticity and lead to impaired cognitive functioning [48,49]. Finally, food insecurity may influence child development through exposure to increased stress and anxiety. For some families, maintaining the household (i.e., energy, water, housing) brings considerable stress and anxiety. Heightened levels of stress and anxiety can impact children and parents physiologically (via triggering the stress-related hypothalamic–pituitary axis), psychologically and socially; including affecting parenting practices and subsequently, child development [47]. As household time and resources are increasingly spent managing food access and availability, the emotional and financial support to facilitate child development may decrease [30]; for example, observed through less money to spend on extracurricular learning/interactive environments. A clear finding of the studies reviewed, was that there are a multitude of variables which are associated with food insecurity and child development outcomes which may protect or amplify the effects of food insecurity. Moreover, food insecurity experienced as worry, or the compromising of food quality and quantity for adults and/or children in a household, has an impact on child development. Child development is also impacted if food insecurity exists for shorter transient or for longer more persistent periods of time. As such, additional factors may need to be considered in exploring the association between food insecurity and child development and using a socio-ecological approach may be key for improving future research.

### 4.2. Applying a Socio-Ecological Lens

When examining the effect of food insecurity on child development it is important to consider multiple risk and protective factors; the socio-ecological model allows us to do this across systems. As part of this review, we categorized the factors taken into consideration by each of the studies as they pertained to each of the systems (Table 3 and Figure 2).

The effects of food insecurity on child development are likely mediated by individual and proximal factors such as the quality of home and school environments, caregiver-child relationships and interactions, parental mental health, and individual differences in biology and temperament [46,47,50]. For example, in this review the important role that maternal mental health, parenting stress and parenting practices played in enhancing or reducing the risks of food insecurity for children’s development was evident [46,47,50]. This was especially apparent for behavioral outcomes. The impact of other caregivers’ mental health [50] and exposure to broader caregiving systems (grandparents and other kinship networks) beyond the immediate home environment tended not to considered. These factors are increasingly recognized as influential on child development outcomes [14].

Distal factors of influence identified included access to social support (borrow money, find emergency childcare) [27] and urbanicity [31]. A number of papers considered societal level factors such as utilizing a food safety net (school meals, SNAP, WIC) [26,27,36], eligibility for social protection measures (TANF, low income energy assistance) [28,30,36], and access to health insurance, which is a pertinent issue in the USA due to the high cost of health care [34,44].

Emerging research in food insecurity is exploring other distal factors that impact on the ability of children to reach their full potential. Recent research has explored broader societal issues, such as lack of social cohesion [51,52], racism [53,54], violence [55], and neighborhood safety [56], and how these impact food insecurity. The links between these factors and child development are well established (see for example [17,57,58,59,60,61]). The next step is exploring the intersection between food insecurity, child development and these concepts.

The socio-ecological framework indicates that children do not operate solely within microenvironments bounded by the household but are influenced by both proximal and distal factors including the broader policy environments influencing food access, availability, affordability, and utilization. Exploring household, family, school, and community environments together will allow a more nuanced picture of the relationship between food insecurity and child development outcomes. This picture will then be able to inform public policy strategies that seek to alleviate poverty and improve the environmental conditions (for example: home, school, community) that contribute to food security, thus influencing child development.

### 4.3. Lessons from the COVID-19 Pandemic

The current ongoing COVID-19 pandemic has highlighted several salient issues regarding food security and child development outcomes. One pertinent issue raised is the fragility of the food systems including food relief and the financial security on which families are dependent. Lockdowns and ongoing uncertainty have increased levels of family stress. This is due to increased demands of balancing childcare/home-schooling/work, financial instability, decreased access to food, and increased incidence of domestic violence [62,63]. The COVID-19 disruption is independently heightening levels of psychological problems, post-traumatic stress symptomology, anxiety, and depression among children [64]. In particular, the pandemic has resulted in childcare and school closures and has highlighted the integral role school food services have in feeding children in families with fragile financial health [65]. The COVID-19 pandemic has shown the myriad of connections and networks across micro-environments and the community that support child development. The effects of the pandemic are yet unrealized but early indications are that consequences will be profound, both in the short and long term.

### 4.4. Limitations

This state-of-the-art review represents food insecurity and child development outcomes over the past decade; however, there are limitations to note. Only papers written in English were reviewed and as such work presented in languages other than English that may represent broader child development outcomes in settings that are not USA-centric may not have been included. We employed broad search terms to capture the food insecurity concept including for example; access, availability, insufficiency. However, given the complexity of the concept, papers that used an alternative term may have been missed. Unlike previous reviews, a majority of papers identified in the last decade used the USDA-HFSSM tool or some variation. This is indicative of the state of current research with the USDA-HFSSM tool being used as a gold standard, however, other tools are available and therefore prior studies using different tools are not represented in the findings of this review. The current review includes a combination of longitudinal and cross-sectional studies and, while longitudinal studies provided stronger evidence of potential pathways through which food insecurity may impact on child developmental outcomes, they do not provide causal pathways.

### 4.5. Future Research Directions

There are several significant issues hindering our ability to determine the effects of food insecurity on child development. These include the inconsistent measurement of and thresholds used to define both food insecurity and child development outcomes. Subsequently, the associations and effects reported are difficult to interpret and our ability to generalize and compare across studies is limited. As such, having assessed the current state of the art literature, we identified the following recommendations and potential opportunities for the future direction of this line of research. These include:Consistent measurement and operationalization of FI including accounting for severity, specifically separating out marginal food security from being fully food secure.High quality studies that explore severity together with persistence/trajectories of FI and its impact on child development.Consistent and standardized measures of child development outcomes.A systematic and socio-ecological (proximal and distal) approach to incorporation of covariates in models.Research conducted beyond the U.S. Given the differences in childcare arrangements, social welfare policies, and practices across countries, the associations between child development and food insecurity needs to be examined in other high-income countries.Exploration of the impact of the COVID-19 pandemic on food security and its influence on child development. The pandemic has been a significant, global event with pervasive socio-cultural consequences that could have long-term impacts on child outcomes.Research that incorporates evidence of children’s diet quality linked to food insecurity severity and persistence and developmental outcomes.Research that asks children about their experiences of food security. To date, only two studies were located [66,67] that asked children directly about their experience.

## 5. Conclusions

This state-of-the-art review indicates that food security status, severity, and persistence do adversely impact upon child development outcomes. The strongest evidence of an effect of food insecurity has been found in academic/cognitive outcomes and externalizing behaviors. The relationship with psychosocial outcomes and internalizing behavior is less clear. Furthermore, longitudinal research on developmental risk and food insecurity is critically needed.

That children in countries producing a surfeit of food are denied the right to quality food is untenable and indicates a failure of political and public will. Furthermore, this situation has likely been exacerbated in recent times with the COVID-19 pandemic, especially in countries where welfare is not easily obtained. The longitudinal socioeconomic effects of this global pandemic are yet to be revealed, but it is foreseeable that there will be significant consequences for ongoing food security, even in many high-income countries, and hence for concurrent and downstream child development outcomes. As such, the time to act is now. What is evident, from this review is that food insecurity is a significant issue in high-income countries. Even if children are not hungry, a level of anxiety about where the next meal is coming from does seem to adversely impact child development. In addition, moving in and out of food security as well as experiencing persistent marginal food security or food insecurity contributes to adverse child development outcomes across cognitive and behavioral domains.

There is an imperative to improve understanding of the association between food insecurity and child development, and further, elucidate the causes of food insecurity. In ameliorating the causes, the right to food can become a reality for all.

## Figures and Tables

**Figure 1 ijerph-18-08990-f001:**
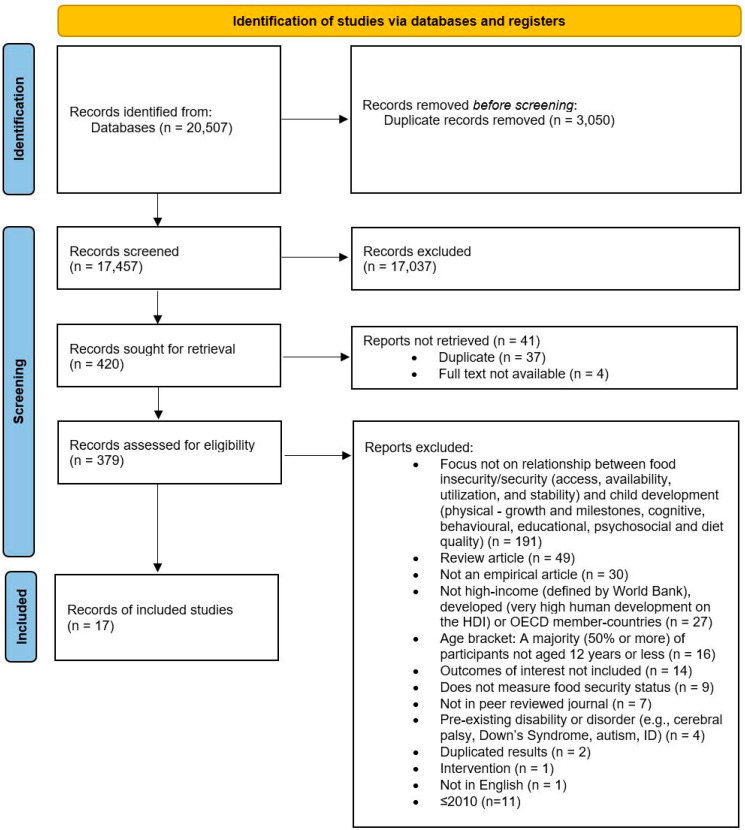
PRISMA diagram of search, adapted from [20].

**Figure 2 ijerph-18-08990-f002:**
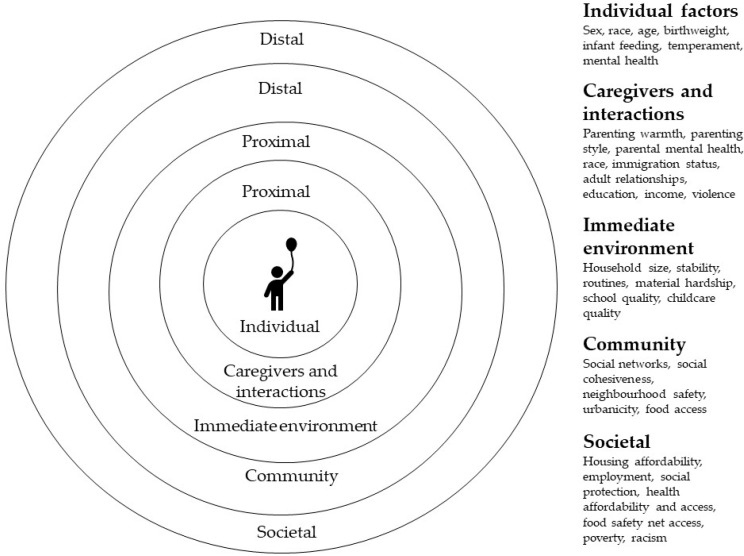
Socio-ecological proximal and distal factors impacting food security and child development (author generated).

**Table 1 ijerph-18-08990-t001:** Inclusion and exclusion criteria.

Criteria	Definition/Rationale
Articles were included from the search if:
≥50% of participants were ≤12 years. Studies were included with children older than the specified age range, only if food insecurity status and developmental outcomes for children aged from birth to 12-years old were identifiable	This group is developmentally distinct from adolescents aged 13–18 years who potentially have more autonomy.
≥50% of participants were living in high income countries	High-income countries have gross national income per capita of $12,736 or more (The World Bank, 2015); developed countries have ‘very high human development’ on the Human Development Index (United Nations Development Programme, 2016); and there are 35 OECD member countries (Organisation for Economic Co-operation and Development, 2017). These lists of countries overlapped substantially, with only one developed country not a high-income or OECD country. In total, 83 countries met these criteria. The search was limited to high income countries due to varying determinants and sequelae of food insecurity in low- and middle-income countries
If they addressed any one of these key child development outcomes: behaviour, cognitive or non-cognitive performance, academic achievement, psychosocial, emotional, developmental risk (e.g., as assessed by PEDS) and motor development	
Articles were excluded from the search if:
A majority of participants were >12 years of age or from low-middle income countries	
They were case studies, letters, commentaries, review articles, conference abstracts with full text unavailable, or theses.	
Did not report on child development outcomes as outlined above but focused on diet quality, nutritional status, weight status, height status, rates of infectious disease, rates of non-communicable disease, mental health diagnoses, hospitalizations	We acknowledge that food insecurity could impact on any or all of the listed factors, and that many of these could mediate or moderate the effect on child development. However, we were interested in the direct impact of food insecurity on child development.
There was no explicit measure of food security.If the same dataset was used and there was a significant overlap in the results. In this instance the older paper with the larger sample size was included.

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
