# Peer review of "Food Insecurity and Child Development: A State-of-the-Art Review"

_ijerph, 2021, doi:10.3390/ijerph18178990_

Round 1
Reviewer 1 Report
The manuscript entitled "Food insecurity and child development: A state-of-the-art review" is a very interesting one. This is a well written paper. The English is good and the data are presented in a clear way. Thus one can quickly read the paper and understand it.
The information presented have been done in a good way, and the data appear to be reliable.
The manuscript presents new and useful data regarding food insecurity that affects children's development. This is a useful addition to the literature. My view is the paper is suitable for publication in International Journal of Environmental Research and Public Health after minor revision as outlined below
Table 2 is missing the footnotes for Y and N.
The numbering of the references is doubled
After the references, the remaining explanations in the template must be deleted
Author Response
Thank you reviewer 1 for your comments.
- Table 2 is missing the footnotes for Y and N. added
- The numbering of the references is doubled Numbers removed
- After the references, the remaining explanations in the template must be deleted template instructions deleted
Reviewer 2 Report
Danielle Gallegos et al. produced an interesting review on the state of the art of food insecurity and child development, taking into account many aspects that could impact on this issue.
The MS is well structured, is easily to understand and it is supported by supplementary material and a rich bibliography.
In order to improve this important MS, please take your precious time to review the following aspects:
- Line 26 abstract: regarding the first appearance in the text of the abstract of "FS", I would cite it in full;
- In paragraph 4 “Discussion - 4.1 Mechanisms of how FI impacts on child development” please mention what are the optimal requirements, provided by the literature in this field and/or by Guidelines, relating to the food intake for the age groups under study, divided by macronutrients. In this paragraph, it could be also useful referring to articles on the impact of the COVID-19 pandemic on food security, if already published, and discussing on this topic;
- Line 350: please cite references in the right way "([27] [39])", similarly for line 352 "([26,36] [27],)";
- Line 423: “That there are children denied” there is probably an error.
Often there are many useless spaces between one word and another, especially after the "dot", so I think it would be appropriate to eliminate what is not necessary, to facilitate the subsequent publication steps (eg. Lines 48, 67, 206, 216, 344, 345).
Lastly, please check again the 63 references according to the instructions for Authors, as Abbreviated Journal Names should be indicated in italics.
Thank you for your efforts in perfecting the article.
Author Response
Dear Reviewer 2 thank you for your comments and considered review. Please find our responses below:
- Line 26 abstract: regarding the first appearance in the text of the abstract of "FS", I would cite it in full; as per reviewer 3 all FI and FS abbreviations have been replaced in full.
- In paragraph 4 “Discussion - 4.1 Mechanisms of how FI impacts on child development” please mention what are the optimal requirements, provided by the literature in this field and/or by Guidelines, relating to the food intake for the age groups under study, divided by macronutrients. Thank you for this recommendation however given that none of the papers looked at diet quality and we have been unable to review this in relation to food security we do not think we should go to this level with identifying the minimum requirements for children. It was important to talk about diet quality as a possible causal mechanism but going into detail will detract from the main point. We have added not discussing diet quality as a limitation of the research.
- In this paragraph, it could be also useful referring to articles on the impact of the COVID-19 pandemic on food security, if already published, and discussing on this topic;
- Line 350: please cite references in the right way "([27] [39])", similarly for line 352 "([26,36] [27],)"; corrected
- Line 423: “That there are children denied” there is probably an error. It was an awkward phrasing – changed to: That children in countries producing a surfeit of food are denied the right to quality food is untenable
Often there are many useless spaces between one word and another, especially after the "dot", so I think it would be appropriate to eliminate what is not necessary, to facilitate the subsequent publication steps (eg. Lines 48, 67, 206, 216, 344, 345). all unnecessary spaces removed
Lastly, please check again the 63 references according to the instructions for Authors, as Abbreviated Journal Names should be indicated in italics. All journal names have been abbreviated and italicised
- Line 26 abstract: regarding the first appearance in the text of the abstract of "FS", I would cite it in full; as per reviewer 3 all FI and FS abbreviations have been replaced in full.
- In paragraph 4 “Discussion - 4.1 Mechanisms of how FI impacts on child development” please mention what are the optimal requirements, provided by the literature in this field and/or by Guidelines, relating to the food intake for the age groups under study, divided by macronutrients.
Thank you for this recommendation however given that none of the papers looked at diet quality and we have been unable to review this in relation to food security we do not think we should go to this level with identifying the minimum requirements for children. It was important to talk about diet quality as a possible causal mechanism but going into detail will detract from the main point. We have added not discussing diet quality as a limitation of the research and a possibility for future research.
Line 472-473 Research that incorporates evidence of children’s diet quality linked to food insecurity severity and persistence and developmental outcomes.
- In this paragraph, it could be also useful referring to articles on the impact of the COVID-19 pandemic on food security, if already published, and discussing on this topic;
Thank you for this insight. We have added a new paragraph as 4.3 in the discussion and as an addition to future research options - this now links with the statement made in the conclusion:
Lines 417-430 4.3 Lessons from the COVID-19 pandemic
The current ongoing COVID-19 pandemic has highlighted several salient issues regarding food security and child development outcomes. One pertinent issue raised is the fragility of the food systems including food relief and the financial security on which families are dependent. Lockdowns and ongoing uncertainty have increased levels of family stress. This is due to increased demands of balancing of childcare/home-schooling/work, financial instability, decreased access to food and increased incidence of domestic violence [63,64]. The COVID-19 disruption is independently heightening levels of psychological problems, post-traumatic stress symptomology, anxiety and depression among children [65]. In particular, the pandemic has resulted in childcare and school closures and has highlighted the integral role school foodservices have in feeding children in families with fragile financial health [66]. The COVID-19 pandemic has shown the myriad of connections and networks across micro-environments and the community that support child development. The effects of the pandemic are yet unrealized but early indications are that consequences will be profound, both in the short and long term.
Lines 468-471 Exploration of the impact of the COVID-19 pandemic on food security and its influence on child development. The pandemic has been a significant, global event with pervasive socio-cultural consequences that could have long-term impacts on child outcomes.
- Line 350: please cite references in the right way "([27] [39])", similarly for line 352 "([26,36] [27],)"; corrected
- Line 423: “That there are children denied” there is probably an error. It was an awkward phrasing – changed to: That children in countries producing a surfeit of food are denied the right to quality food is untenable
Often there are many useless spaces between one word and another, especially after the "dot", so I think it would be appropriate to eliminate what is not necessary, to facilitate the subsequent publication steps (eg. Lines 48, 67, 206, 216, 344, 345). all unnecessary spaces removed
Lastly, please check again the 63 references according to the instructions for Authors, as Abbreviated Journal Names should be indicated in italics. All journal names have been abbreviated and italicised
Reviewer 3 Report
The paper “Food insecurity and child development: A state-of-the art review” deals with very interesting and important topic.
Having said that, your study is very promising, but it needs some improvements. With the intention that you can make those improvements in your future research, I will comment on some of the most relevant weaknesses found in the different sections. I hope you find those comments useful.
Please replace the abbreviation FI and FS with the full name food insecurity, food security throughout the text of the paper. We use abbreviations when it is a common shorthand e.g. European Union - EU.
Remove unnecessary spaces, e.g. line: 48, 67, 206, 228 etc.
Data sources under figures and tables are missing.
In discussion Authors should also clearly present the following important aspects:
- the research limitations;
- the future research directions.
Please prepare a bibliography according to the journal's guidelines for authors (the name of the journal should be written in italics).
Remove double numbering from the bibliography.
I hope these comments can be of help in improving the paper and encourage the authors to move forward.
Author Response
Dear Reviewer 3
Thank you for your insightful comments and suggestions.
- Please replace the abbreviation FI and FS with the full name food insecurity, food security throughout the text of the paper. We use abbreviations when it is a common shorthand e.g. European Union - EU. All abbreviations converted to full text
- Remove unnecessary spaces, e.g. line: 48, 67, 206, 228 etc. all unnecessary spaces removed
- Data sources under figures and tables are missing. Data sources for the tables are included next to each author name but were missing for Table 3 which have now been added. The Figure 2 was author generated and so this has been put in brackets. Data source for Figure 1 has been added
- In discussion Authors should also clearly present the following important aspects:
- - the research limitations; we have added a limitations section
Lines 433-447 This state-of-the-art review represents food insecurity and child development outcomes over the past decade, however there are a number of limitations to note. Only papers written in English were reviewed and as such work presented in languages other than English that may represent broader child development outcomes in settings that are not USA-centric may not have been included. We employed broad search terms to capture the food insecurity concept including for example; access, availability, insufficiency. However, given the complexity of the concept, papers that used an alternative term may have been missed. Unlike previous reviews, a majority of papers identified in the last decade used the USDA-HFSSM tool or some variation. This is indicative of the state of current research with the USDA-HFSSM tool being used as a gold standard, however, other tools are available and therefore prior studies using different tools are not represented in the findings of this review The current review includes a combination of longitudinal and cross-sectional studies and, while longitudinal studies provided stronger, evidence of potential pathways through which food insecurity may impact on on child developmental outcomes, they do not provide causal pathways.
- the future research directions. This is included under a separate heading we have now renamed this “future research directions”
Please prepare a bibliography according to the journal's guidelines for authors (the name of the journal should be written in italics). The bibliography has now been prepared in this way
Remove double numbering from the bibliography. removed
Round 2
Reviewer 2 Report
Well done, thanks for taking my suggestions.The manuscript is very admirable, I suggest to correctly change the numbering of paragraph 4.3, as it is a duplicate ("lessons from the COVID-19 pandemic" and "limitations").
Author Response
Thank you so much for picking this up
4.3 Limitations - changed to 4.4
4.4 Future Research directions - changed to 4.5
